# Closed-Loop Microbial Fuel Cell Control System Designed for Online Monitoring of TOC Dynamic Characteristics in Public Swimming Pool

**DOI:** 10.3390/ijerph192013024

**Published:** 2022-10-11

**Authors:** Haishan Chen, Xiaoping Meng, Dianlei Liu, Wei Wang, Xiaodong Xing, Zhiyong Zhang, Chen Dong

**Affiliations:** 1Department of Health Service and Management, School of Sport Management, Shandong Sport University, Jinan 250102, China; 2Institute of Environment and Ecology, Tsinghua Shenzhen International Graduate School, Tsinghua University, Shenzhen 518055, China; 3Human Science Center, Ludwig-Maximilians-Universität, 280539 Munich, Germany

**Keywords:** public swimming pool, total organic carbon, closed-loop MFC-biosensor control system, computer simulation

## Abstract

Total organic carbon (TOC) in the water of public swimming pools (PSPs) must be monitored online for public health. In order to address the shortcomings of conventional microbial fuel cell biosensor (MFC-biosensor), an innovative biosensor with peculiar closed-loop structure was developed for online monitoring of TOC in PSPs. Its design was based on experimental data, model identification, cybernetics, and digital and real-time simulation. The outcomes of the digital simulation demonstrated that the closed-loop MFC control system possesses the desired structure with a pair of dominant complex-conjugate closed-loop poles (−15.47 ± 7.73j), and the real-time simulation showed that its controller output signals can automatically and precisely track the variation in TOC concentration in PSP water with the desired dynamic response performances; for example, mean delay time was 0.06 h, rise time was 0.12 h, peak time was 0.18 h, maximum overshoot was 7.39%, settling time was 0.22 h, and best fit 0.98. The proposed principle and method of the closed-loop MFC-biosensor control system in the article can also be applied for online monitoring of other substances in water, such as heavy metal ions, chemical toxicants, and so forth, and lay a theoretical foundation for MFC-based online monitoring substances in an aquatic environment.

## 1. Introduction

The organic contaminations in public swimming pool (PSP) water can cause serious harm to the health of swimmers, such as skin disease, allergies, enteric disease, and even certain cancers along with constant accumulation of the organic pollutants (such as urea, chlorourea, and so on) in PSP water, which has received extensive attention [1,2,3]. There is an urgent need to find a cheaper and more effective way for online monitoring of the organic matters concentration in the PSP water, in order to rapidly implement control measures to keep these organic pollutants at acceptable levels.

In the research, total organic carbon (TOC) is used as the index to indicate the content of organic matter in the PSP water, because the TOC can reflect the degree of organic matters contamination in the PSP water more comprehensively compared with other indexes, such as urea index and chemical oxygen demand (COD). At present, water sampling from the PSP and then traditional offline TOC measurement is extensively used for water quality monitoring. However, offline measurement methods with high cost and slow speed cannot capture the complicated dynamic response characteristics of water pollutants at all, and provide real-time warning as TOC exceeds standard.

During the past decade, there is an increasing interest in the research of microbial fuel cell biosensors (MFC-biosensors). With a simple construction and no additional transducer, MFC-biosensors are applicable for the in situ and online monitoring of water quality [4,5,6]. MFCs use electricigens as biocatalysts to convert chemical energy in organic matters to electricity [7]. Furthermore, the development of new fuel cells has been rapid and the application area has been extended in recent years [8,9,10].

This novel bioelectrochemical technology makes it feasible to achieve improvement of wastewater treatment efficiency and recovery of energy simultaneously [11]. Theoretically, as substrates of electricigens, the MFC-biosensors output voltages can reflect organic matters concentration in the PSP directly based on their quantitative relation determined by linear regression; therefore, MFC-biosensors are extensively applied for open-loop prediction for water pollutants [12,13,14]. However, open-loop predictions have two obvious disadvantages as follows:(1)Poor stability. As is known, the open-loop prediction of water pollutants based on output voltages of MFC-biosensors is very sensitive to internal variations and external disturbances, frequently causing the large prediction errors of water pollutants concentration [15].(2)Transient process of pollutant variation cannot be tracked and captured. According to cybernetics, the open-loop prediction must acquire the steady-state value of water pollutants concentration. However, it often takes quite long time for water pollutants concentration to be stabilized, and concentrations exceeding permissible standards may occur in the transient process of pollutants concentrations variation, which cannot be monitored at all [16].

In order to fill the above-mentioned gap, in the article, the open-loop MFC-biosensor was designed as the closed-loop structure with proper feedback controller. Based on the viewpoints of cybernetics, the closed-loop MFC-biosensor control system possesses a good capability against internal variations and external disturbances to effectively maintain stability operation and prediction accuracy of MFC-biosensor, and simultaneously, it can also accurately predict and track the pollutants dynamic process including the whole transient and steady-state response. Beyond this, the closed-loop structure can effectively reduce the effect of nonlinearities such as dead zone, backlash, and Coulomb friction as well as the kinetic model of MFC-biosensor being somewhat inaccurate in the practical application. In the article, the results demonstrated that a particular closed-loop MFC biosensor control system constructed and built from kinetic model identification and digital and real-time simulation can timely and precisely track the dynamic processes of TOC concentration in the PSP water with desired dynamic response performance.

Based on the deficiencies of current MFC-biosensors, in the article, an innovative closed-loop MFC-biosensor structure with proper feedback controller for monitoring the dynamic process (including transient and steady-state process) of TOC concentration in the PSP water was successfully designed and built through experimental data, system identification, cybernetics, and digital and real-time simulation. The results demonstrated that the closed-loop MFC biosensor control system can timely and precisely track the change process of TOC concentration in the PSP water with desired dynamic response performance. This research could also lay a theoretical and application foundation to design and build performance-enhanced MFC-biosensors for online monitoring other organic pollutants and even toxins in water, providing valuable information for effective prevention and control of water pollutions.

## 2. Materials and Methods

### 2.1. Prototype of MFC-Biosensor for Online Monitoring TOC Concentration

A MFC with dual-chamber configuration was constructed as biosensor for online monitoring TOC concentration in PSP water (Figure 1). Three groups of dual-chamber MFC were constructed, in which the anode and cathode compartments were 25 mL and 42 mL, respectively. The two chambers were separated by a proton exchange membrane (Nafion 117, Hesen Electric Corporation, China). The anode was 3 cm × 2.5 cm carbon cloth (HCP330, Hesen Electric Corporation, China), which was soaked in acetone overnight before use, then dried and subjected to high temperature ammonification. The cathode was 2.0 cm × 2.0 cm platinum-loaded carbon paper with a platinum-loaded capacity of 0.5 mg/cm^2^. The external resistance of the MFC load is 330 Ω. The anode inoculation source was the anode effluent from MFC that had been running stably for more than half a year in the laboratory, and the electrochemically active microorganism of the mixed strains was mainly *Geobacter*. After inoculation, the MFC of each group was placed in a constant temperature incubator at 27.0 ± 0.5 °C, and the anode chamber was self-circulated at 3 mL/min through a peristaltic pump to accelerate mass transfer and promote the formation of biofilms. Each MFC was connected to the data acquisition system to measure the output voltage of the MFC in real time. When the maximum voltage does not increase and three cycles of repeatable output voltage are detected, the MFC startup is considered to be complete. The fresh anolyte used consisted of 5.85 g NaCl, 0.13 g KCl, 0.31 g NH_4_Cl, 6.08 g NaH_2_PO_4_·2H_2_O, 21.83 g Na_2_HPO_4_·12H_2_O, 12.50 mL trace minerals solution, and 5.00 mL vitamin solution in 1.00 L deionized water. The catholyte consisted of 5.85 g NaCl, 6.08 g NaH_2_PO_4_·2H_2_O, and 21.83 g Na_2_HPO_4_·12H_2_O in 1.00 L deionized water [17,18]. All the experiments were conducted at a temperature of 27.0 ± 0.5 °C in accordance with PSP water. During the operation of MFC, TOC in PSP water was used as an organic substrate injected into the anolyte through an inflow pipe.

### 2.2. Construction of Closed-Loop MFC-Biosensor for Online Monitoring TOC Concentration in PSP Water through Digital Simulation

#### 2.2.1. Data Acquisition

The water samplings were obtained from PSP in Shandong Sport University, and TOC concentration was analyzed by Elementar liqui TOC Ⅱ (German, Elementar) whose measurement range was from 0 to 20 mg/L. According to the input of TOC concentrations in PSP water, the MFC-biosensor transmitted them in the form of output voltages whose magnitude ranged from 0 V to 0.5 V. Hence, the time-series input-output (IO) data were obtained for kinetic model identification and validation as well as design and optimization of closed-loop MFC-biosensor for monitoring TOC concentration in PSP water through digital simulation.

#### 2.2.2. Model Identification and Construction of Closed-Loop MFC-Biosensor Control System

Based on time-series IO data, a kinetic model of MFC-biosensor transfer function composed of static gain, process pole and zero, and response delay was identified on the platform of Matlab/System identification toolbox, and then its feedback controller was designed in both time- and frequency-domain based on cybernetics and digital simulation. The closed-loop MFC-biosensor control system was constructed through feedback connection between the feedback controller and transfer function of MFC-biosensor to ensure the output voltages perfectly in agreement with set reference signals, even if the transfer function of MFC-biosensor is somewhat inaccurate, or with the existence of internal variations and external disturbances.

### 2.3. Rapid Prototyping of the Closed-Loop MFC-Biosensor Control System for Monitoring TOC Concentration in PSP Water by Real-Time Simulation

After the closed-loop MFC-biosensor control system for monitoring TOC concentration dynamic process in PSP water was designed and optimized through digital simulation, the transfer function of feedback controller and MFC-biosensor were rapidly prototyped by C code generation and embedded into a chip which was feedback connected with the prototype of MFC-biosensor. Such hardware-in-loop (HIL) structure was used to carry out real-time simulation for testing monitoring efficacy practically. Based on the actual effect of monitoring TOC concentration in PSP water, the feedback controller and MFC-biosensor transfer function were further optimized and calibrated, respectively [19,20].

## 3. Results and Discussion

### 3.1. Model Identification between TOC Concentration and MFC-Biosensor Output Voltage

#### 3.1.1. Data Preprocessing

Before the transfer function between TOC concentrations and output voltages of MFC-biosensor was identified from time-series IO data, the outliers were removed. The time-series IO data located out of the range between μ (average) − 3σ (standard) and μ + 3σ were considered as outliers and excluded. These outliers might be caused by signal spikes or by measurement malfunctions during experimental data acquisition, and adversely affect the model identification. Simultaneously, the means, offsets, linear trends, and high-frequency noise ranging from 500 rad/s to 1300 rad/s were also removed from the regularly sampled time-series IO data in order to obtain more accurate kinetic model.

Because the off-line sampling and analysis period of TOC concentration from PSP is relatively long, a small amount of time-series IO data might not make model identification process convergent; therefore, the B-spline interpolation was applied for accurate estimation of TOC concentration values at non-sampling time instants to obtain enough time-series IO data for model identification (Figure 2).

After data preprocessing, these time-series IO data were divided into two parts, half of them were applied for model identification and the other half for model validation.

#### 3.1.2. Model Identification and Validation from Time-Series IO Data

(1)Model identification

The microbial community in MFC-biosensor could be considered to stay in normal state according to practical operation of MFC-biosensor in PSP, signifying that the richness and abundance of microbial community would not have a significant change in the case of internal variations and external disturbances; therefore. the kinetic model of MFC-biosensor could be supposedly unchangeable in the course of online monitoring of TOC in PSP, which was the prerequisite to develop the kinetic model of MFC-biosensor.

Based on time-series IO data between TOC concentrations and MFC-biosensor output voltages (Figure 2), a simplest process model was identified via trial-and-error methods on the platform of MATLAB/System Identification Toolbox as follow:(1)$$\frac{V\left(s\right)}{\mathrm{TOC}\left(s\right)}=\frac{0.02\left(1+1.07s\right)}{\;\left(1+3.66s\right)\;\left(1+0.51s\right)}e^{-0.2s}$$
where *s* is the Laplace operator. As shown in Equation (1), however, a relatively long time-delay (Td = 0.2 h) of MFC-biosensor output voltage responding to the TOC concentration variation, which reflects electrical generation characteristics of microbial community, and has adverse effect on the design and construction of the closed-loop MFC control system. Hence, time delay was substituted by a rational model of transfer function with minimal orders through Padé approximation, and therefore Equation (1) can be approximately expressed as follows:(2)$$\frac{V\left(s\right)}{\mathrm{TOC}\left(s\right)}\approx \frac{0.02s^{3}-0.59s^{2}+5.57s+5.77}{1.85s^{4}+60.04s^{3}+688s^{2}+1293s+303}$$

(2)Model validation

The three common indexes, best fit (*BF*), Spearman correlation (*r*) coefficient, and Nash–Sutcliffe efficiency standard deviation (*SD*) were applied for model validation by measurement of similarity between measured output voltage of MFC-biosensor in experiment and prediction from its transfer function (Equation (2)).

The best fit and Nash–Sutcliffe efficiency standard deviation are respectively defined as follows:(3)$$BF=\left(1-\frac{\left|y-y_{p}\right|}{\left|y-y_{m}\right|}\right)$$
(4)$$SD=1-\left[\frac{\sum_{i=1}^{n}{\left(y-y_{p}\right)}^{2}}{\sum_{i=1}^{n}{\left(y-y_{m}\right)}^{2}}\right]$$
where *y* is the measured output, *y_p_* is the simulated output, and *y_m_* is the mean of *y*. Both *BF* and *SD* are between 0 and 1, 1 corresponds to a perfect fit, and 0 indicates that the fit is no better than guessing the output to be a constant, i.e., *y_p_* = *y_m_*.

From Figure 3, the transfer function of MFC-biosensor is highly valid and can be applied for design and optimization of a closed-loop MFC-biosensor control system for online monitoring of TOC concentration in PSP water.

### 3.2. Design and Optimization of the Closed-Loop MFC-Biosensor Control System

Although the transfer function (Equation (2)) between TOC concentration and output voltage is stable due to all poles locating on the left hand of complex plane, it cannot be applied for online monitoring of TOC concentration in PSP water in an open-loop prediction mode, since the position of zeros and poles in complex plane are not satisfied at all, which will result in undesirable transient responses to TOC concentration variations in PSP water.

Therefore, the closed-loop MFC-biosensor control system was designed by root locus method in time-domain and by Bode diagram in frequency-domain in combination with Ziegler–Nichols rules for tuning feedback controller (Figure 4), and the optimal closed-loop MFC-biosensor was finally designed and constructed with the desired feedback controller with 2 zeros and 2 poles successfully obtained as follows:(5)$$C=\frac{70.22\left(s+0.70\right)\left(s+2.8\right)}{\left(s+3.5\right)\left(s+4.5\right)}$$
which could precisely track different reference control inputs with desired dynamic responses characteristics including transient and steady-state response characteristics [21]. It is worth mentioning that the optimal closed-loop MFC-biosensor control system has a pair of dominant complex-conjugate closed-loop poles (−15.47 ± 7.73j), so that the response of the system is dominated by this pair of complex-conjugate closed-loop poles. The presence of such poles can effectively reduce the effect of nonlinearities such as dead zone, backlash, and Coulomb friction, and even the transfer function of MFC-biosensor is somewhat inaccurate due to internal variations and external variations in the practical application, realizing tract accurately, and predicting the dynamic process of TOC in PSP [22]. Simultaneously, the optimal closed-loop MFC-biosensor control system has also satisfied phase margin and gain margin to produce the best dynamic response performance with proper delay time, rise time, peak time, overshoot, and settling time.

Based on the transfer function of MFC-biosensor and its feedback controller, the simulation model of the closed-loop MFC-biosensor was established on the platform of Matlab/Simulink to investigate its dynamic response characteristics to different reference voltage inputs. Based on cybernetics, if the closed-loop MFC-biosensor control system can perfectly track the reference inputs, the output of feedback controller is exactly the prediction of TOC concentration in the PSP water. In order to prove it, the simulation model of closed-loop MFC control system was changed by inserting a copy of MFC-biosensor between the signal generator block producing different reference inputs of actual TOC concentration in PSP water and the *sum* block (Figure 5).

As illustrated in Figure 5, the *P2DZ* block is MFC-biosensor’s transfer function with 2 poles, 1 zero and 1 time delay (Equation (1)), the *Copy of MFC-biosensor* block is a clone of the *P2DZ* block, the *C* block is feedback controller’s transfer function (Equation (5)), the *Mux* block combines the actual TOC concentration and feedback controller output into a single vector displayed in a *Scope* block, the *Sum* block performs subtraction on its inputs which are different outputs of two same *MFC-biosensor* blocks, and the *Actual TOC Concentration* block is a signal generator producing different actual TOC concentrations signals used as reference inputs. From the viewpoint of signals and systems science, the common physical signals may be considered as step signals, square-wave signal, ramp signals, parabolic signals, sinusoidal signals, or stochastic signals, as well as their combinations [22]. Hence, these common signals were generated as actual TOC in PSP water, i.e., reference inputs, to numerically test the dynamic response characteristics of the closed-loop MFC-sensor control system. As illustrated in Figure 6, the results of the large number of digital simulations proved that the closed-loop MFC-sensor control system possessed high accuracy with good dynamic performance including precise transient and steady-state response characteristics to these common physical signals and their combinations.

### 3.3. Real-Time Simulation of the Closed-Loop MFC-Biosensor Control System for Online Monitoring TOC Concentration in PSP Water

After obtaining the simulation model of the optimal closed-loop MFC-biosensor control system through cybernetics and digital simulation, the simulation model of MFC-biosensor in the closed-loop structure, as illustrated in Figure 5, was replaced by MFC-biosensor prototype to form HIL structure, carrying out real-time simulation to practically test the monitoring efficacy and the validation of modeling, design, optimization [20,23]. Then, the rapid prototyping of the feedback controller and the copy of the simulation model of MFC-biosensor, as illustrated in Figure 5, were conducted by the C code generation of their simulation models for building to executable programs, and these codes were embedded in a chip as actual prototype of controller for online monitoring of TOC concentration in PSP water on the platform of MatLab/Real-Time Workshop (RTW). Aside from MFC-biosensor prototype, the chip operating the C code of simulation models of the feedback controller and the copy of the simulation model of MFC-biosensor, the HIL structure also included signal amplifier, signal indicator, data acquisition card (USB-1608FS, Omega, Norwalk, CT, USA), graphical user interface (Figure 7a,b), and other peripheral circuits as well as related bundled software for real-time sampling and recording MFC-biosensor’s output voltage response to TOC concentration variation in PSP water.

A real-time simulation was conducted to practically test the closed-loop MFC-biosensor control system for monitoring efficacy of TOC concentration dynamic process in PSP of Shandong Sport University, the volume of PSP was 1500 m^3^ with 50 m long, 20 m wide, 1.5 m high. The distilled water was firstly poured into PSP, and then urea or the distilled water was added randomly every 2 h to PSP water, which was simultaneously stirred constantly for even distribution of TOC concentration. Therefore, TOC concentration in PSP water presented stepwise variations which can be precisely obtained through theoretical calculation in combination with sampling analysis. Measurement of TOC concentration was carried out every 5 min, which was a reasonably short sampling time for the monitoring of TOC concentration dynamic process in PSP water. As illustrated in Figure 8, the monitoring results clearly verified the validity of the proposed theory and method through design and building of an optimal closed-loop MFC control system for online monitoring TOC concentration in PSP water.

Specifically, the dynamic response characteristics of the practical closed-loop MFC-biosensor control system for online monitoring TOC were specified by six indexes as follows:(1)Delay time (t_d_) is the time needed for the response to reach half the final value, i.e., state-steady value (S_v_), which is the prediction of closed-loop MFC-biosensor control system behaves as time approaches infinity.(2)Rise time (t_r_) is the time required for the response to rise from 5% to 95% of S_v_.(3)Peak time (t_p_) is the time required for the response to reach the first peak of the overshoot.(4)Maximum overshoot (M_p_) is the maximum peak value of the response curve, the amount of the Mp directly indicates the relative stability of the dynamic system.(5)Settling time (t_s_) is the time required for the response curve to reach and stay within 2% of the final value, i.e., state-steady value.(6)Best fit (*BF*) between actual TOC concentration and S_v_.

Based on experimental data and prediction data of closed-loop MFC-biosensor control system (Figure 8), the average of above specification was calculated, i.e., t_d_ was 0.06 h, t_r_ was 0.12 h, t_p_ was 0.18 h, M_p_ was 7.39%, t_s_ was 0.22 h, and *BF* was 0.98.

It is worth mentioning that the traditional MFC-biosensor is designed as a particular closed-loop control system for robust prediction of TOC concentration in PSP. According to cybernetics, feedback controller can compensate capability loss in TOC concentration prediction under internal variations and external disturbances and eliminate their adverse influence on TOC concentration prediction to maintain desired predication performance of closed-loop MFC-biosensor control system.

In addition, the principle and method proposed in the article could also be applied for online monitoring toxic substances in aquatic environment, because some toxic substances such as heavy metals and chemicals must influence electricity generation performance of microbial community [18]; therefore, the kinetical model describing relationship between toxic substances concentration and microbial electricity generation characteristics could also be established and corresponding closed-loop MFC-biosensor control system could also be constructed for online monitoring these toxic substances in aquatic environments in a real-time, precise, and reliable way.

Although a transfer function was identified from experimental data and used for development of constitutive relationship between TOC concentration variations in PSP water and the output voltage dynamic responses of the MFC-biosensor, the relationship between concentration of toxic substances (such as heavy metals and chemicals) and output voltage of MFC-biosensor sometimes might be highly nonlinear. This is because the heavy metal ions and chemical toxicants will lead to more serious damage to microbial community in MFC-biosensor, the strong nonlinear dynamic characteristics must be produced in the process of electricity production.

Hence, in the future, a nonlinear kinetic model of MFC-biosensor must be identified from the experimental data to effectively capture the nonlinear response characteristics of MFC-biosensor voltage to toxic substances concentration variations in the aquatic environment, the nonlinear kinetic models of MFC-biosensor could be identified by nontraditional methods, such as fuzz inference system (FIS), artificial neural networks (ANN) [24,25,26], and simultaneously corresponding feedback nonlinear controller should be developed by theories and methods of modern cybernetics, such as robust control, self-adaptive control, fuzzy logic control, artificial neural networks, and so on [27,28]. The upgraded closed-loop MFC-biosensor control systems with nonlinear kinetic model and feedback controller also have proposed structure as shown in Figure 5.

## 4. Conclusions

In the article, a specific MFC was used as a specific biosensor for online monitoring TOC variations in PSP water. This MFC-biosensor was successfully designed as an optimal closed-loop control system based on experimental data, kinetic model identification, cybernetics, and digital and real-time simulation. The results proved that the output signals of the feedback controller of the closed-loop MFC-biosensor control system can precisely and robustly track the TOC concentration dynamic process in PSP water with desired dynamic response characteristics. This research addressed two of the current MFC-biosensor’s shortcomings and lay a theoretical and methodological foundation to design and build an advanced MFC-biosensor for online monitoring of other substances in aquatic environments.

## Figures and Tables

**Figure 1 ijerph-19-13024-f001:**
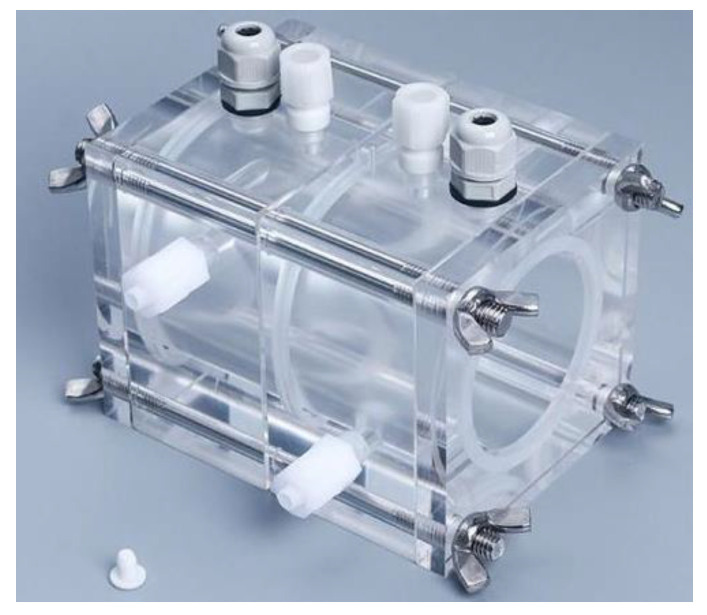
A unit of MFC prototype.

**Figure 2 ijerph-19-13024-f002:**
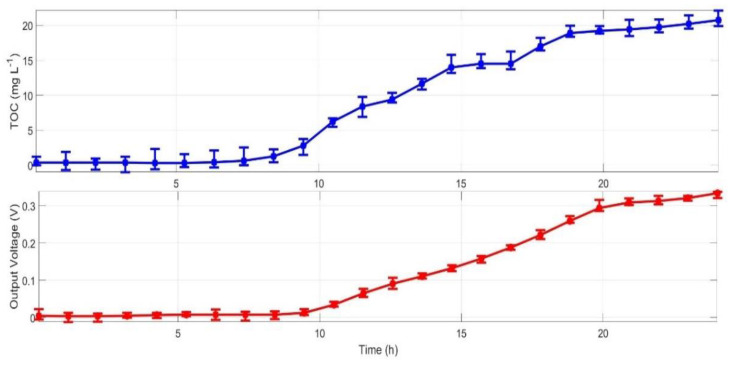
Time-series IO data between TOC concentration and output voltage of MFC-biosensor via data preprocessing.

**Figure 3 ijerph-19-13024-f003:**
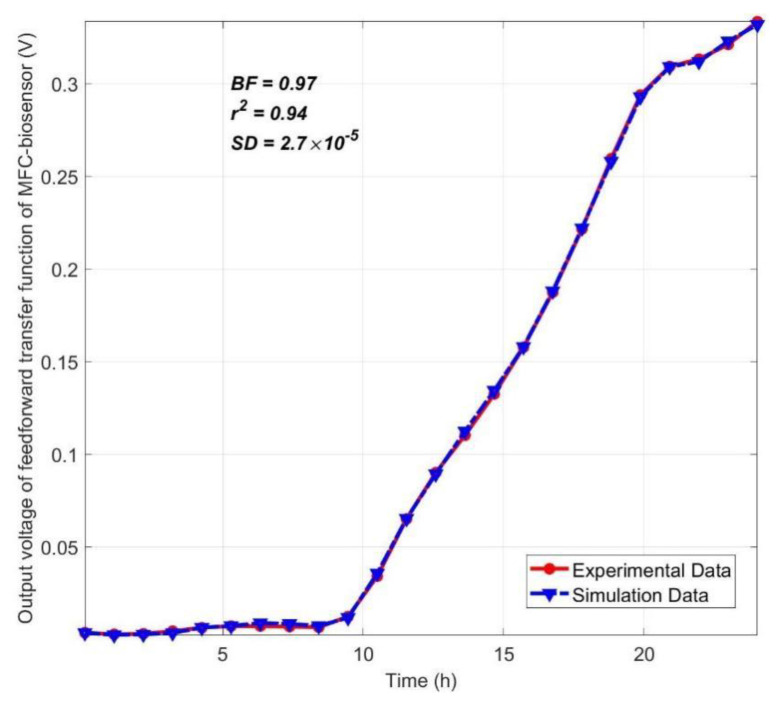
Model validation of transfer function of MFC-biosensor.

**Figure 4 ijerph-19-13024-f004:**
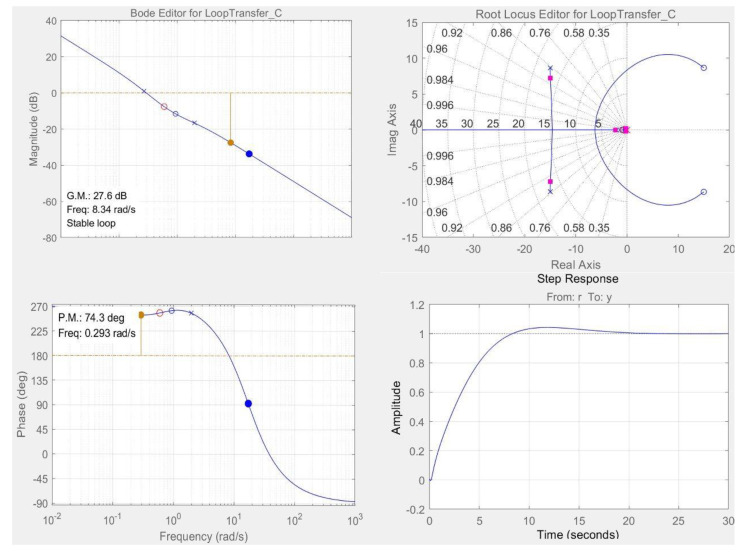
Design of closed-loop MFC control system by root locus method in time domain and Bode diagram in frequency domain.

**Figure 5 ijerph-19-13024-f005:**
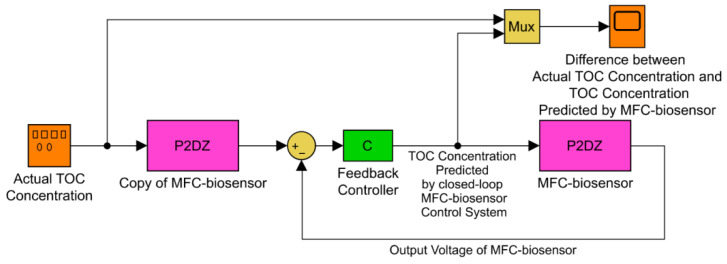
Simulation model of the closed-loop MFC-biosensor control system for online monitoring TOC concentration in PSP water.

**Figure 6 ijerph-19-13024-f006:**
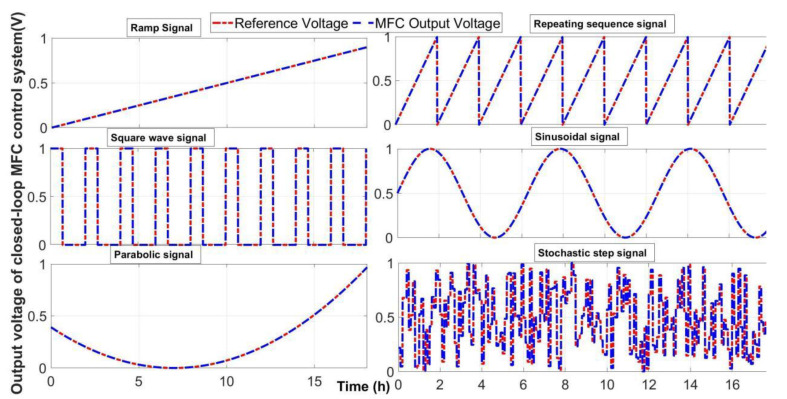
TOC prediction from the feedback controller of the closed-loop MFC-biosensor control system with different TOC concentrations in PSP water as reference inputs.

**Figure 7 ijerph-19-13024-f007:**
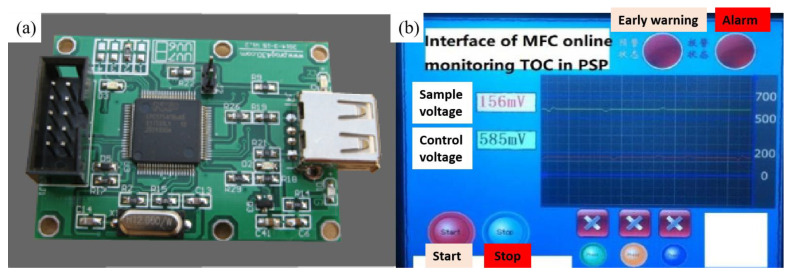
Data acquisition card (**a**) and graphical user interface of online monitoring TOC in PSP by the closed-loop MFC-biosensor control system (**b**).

**Figure 8 ijerph-19-13024-f008:**
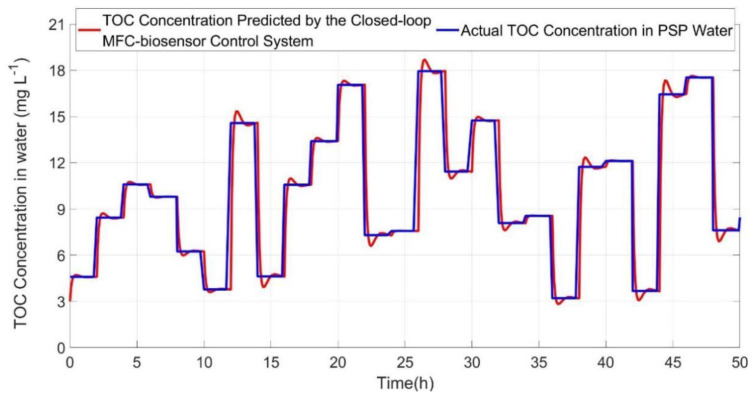
Real-time simulation result of the closed-loop MFC-biosensor control system for monitoring TOC concentration in PSP of Shandong Sport University.

## Data Availability

Not applicable.

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
