# Peer review of "Closed-Loop Microbial Fuel Cell Control System Designed for Online Monitoring of TOC Dynamic Characteristics in Public Swimming Pool"

_ijerph, 2022, doi:10.3390/ijerph192013024_

Round 1

Reviewer 1 Report

This work employs an innovative microbial fuel cell biosensor (MFC-biosensor) was developed for online moni-13 toring TOC in PSP, which is based on experimental data, model identification, cybernetics, 14 digital and real-time simulation. The following are some observations:

1. All abbreviations should be spelled out at the first occurrence in abstract and introduction.

2. “experimental data” can be found both in line 14 and 15, please correct it.

3. Fig. 4 is not clearly presented. The quality of the figure should be improved.

4. The validation of the model is should be described more details.

5. In the section of introduction, I think there are other related references in this research field as follows

Multi-Sub-Inlets at Cathode Flow-Field Plate for Current Density Homogenization and Enhancement of PEM Fuel Cells in Low Relative Humidity. Energy Conversion and Management.2022, 252, 115069

6. Sections 3 and 4 should merge into one section: results and discussions.

7. It is better to provide some quantitative results in the abstract and conclusion.

Author Response

Response to reviewer 1

We would like to thank the reviewers for the valuable comments and constructive suggestions that have improved the quality and clarity of the manuscript. Revised parts are marked in the manuscript.

This work employs an innovative microbial fuel cell biosensor (MFC-biosensor) was developed for online moni-13 toring TOC in PSP, which is based on experimental data, model identification, cybernetics, 14 digital and real-time simulation. The following are some observations:

1.All abbreviations should be spelled out at the first occurrence in abstract and introduction.

Thank you very much for your reminding. You can find the revised abbreviation part in the new version.

2.“experimental data” can be found both in line 14 and 15, please correct it.

This suggestion is quite reasonable. We have deleted one in the revised manuscript.

  1. Fig. 4 is not clearly presented. The quality of the figure should be improved.

Thank you for this good advice. We have exchanged new figure into the revised paper. The quality should be better.

  1. The validation of the model is should be described more details.

Thanks for the good advice. We used three indicators, best fit (BF), Spearman correlation (r) coefficient and standard deviation (SD), to test the effectiveness of the model. From the perspective of behavioral similarity, They adequately demonstrate the high degree of accuracy of the model. According to the request of the reviewer, we added some contents to the revised manuscript to further explain the rationality of the validity test.

  1. In the section of introduction, I think there are other related references in this research field as follows

Multi-Sub-Inlets at Cathode Flow-Field Plate for Current Density Homogenization and Enhancement of PEM Fuel Cells in Low Relative Humidity. Energy Conversion and Management.2022, 252, 115069

Thank you for this good advice. We have added it and others into the revised paper.

References:

Multi-Sub-Inlets at Cathode Flow-Field Plate for Current Density Homogenization and Enhancement of PEM Fuel Cells in Low Relative Humidity. Energy Conversion and Management, 2022, 252: 115069

Three-dimensional numerical study of a cathode gas diffusion layer with a through/in plane synergetic gradient porosity distribution for PEM fuel cells. International Journal of Heat and Mass Transfer, 2022: 188, 122661.

Lattice Boltzmann simulation of a gas diffusion layer with a gradient polytetrafluoroethylene distribution for a proton exchange membrane fuel cell. Applied Energy, 2022, 320:119248.

6.Sections 3 and 4 should merge into one section: results and discussions.

Thank you very much for your positive and valuable comments. The new version was merged with results and discussions.

7.It is better to provide some quantitative results in the abstract and conclusion.

We have provided more detailed description of these in revised manuscript, and think it could ensure the accuracy and the reliability of our study.

Reviewer 2 Report

This research proposed a closed-loop MFC control system for online tracking TOC in the public swimming pool. The idea and result are quite interesting. However, the following points should be improved.

(1)  In the Introduction, the authors pointed out that open-loop prediction systems have "poor stability," and the authors claimed that the proposed system could improve that backward. How the proposed system deal with this point may not have been described clearly in the manuscript.

(2)  How was the stability of the system? Since only 50 h long data was presented.

(3)  Error of the proposed system has not been discussed related to Fig. 8. 

(4)  More recent research should be cited ( the latest was from 2018) to show the related trends and novelty of this research.

(5)  More information related to Fig. 5 should be provided. For example, how was ‘copy of MFC-biosensor’ made? What was ‘Mux’?

Author Response

Response to reviewer 2

We would like to thank the reviewers for the valuable comments and constructive suggestions that have improved the quality and clarity of the manuscript. Revised parts are marked in the manuscript.

This research proposed a closed-loop MFC control system for online tracking TOC in the public swimming pool. The idea and result are quite interesting. However, the following points should be improved.

1.In the Introduction, the authors pointed out that open-loop prediction systems have "poor stability," and the authors claimed that the proposed system could improve that backward. How the proposed system deal with this point may not have been described clearly in the manuscript.

The direct prediction of TOC in PSP based on MFC is an open-loop prediction system. The control principle tells us that the open-loop structure is very vulnerable to internal variations and external disturbances. We innovatively use cybernetics to design the open-loop prediction structure of MFC into a closed-loop control structure. The proposed method has been discussed in this paper. In order to highlight this innovation, we have added relevant content in the revised manuscript.

2.How was the stability of the system? Since only 50 h long data was presented.

The stability refers to the MFC is the stability of the closed-loop control system, including the stability of the structure-- the location of the closed-loop poles (the absolute stability), and the stability of the tracking performance (relative stability). We adopted the root locus method in time-domain and Bode diagram method in frequency-domain. The closed-loop control system of the MFC poles are on the left half complex plane. And it has a pair of dominant complex-conjugate closed-loop poles. For the monitoring of TOC in PSP, we believe that 50 h is a long enough time, and FIG. 8 shows that the closed-loop MFC control system has a good dynamic prediction performance.

3.Error of the proposed system has not been discussed related to Fig. 8.

Based on cybernetics, in this paper, this error of the proposed system refers to the error between the actual output (Red Line in Fig. 8) and the real-time tracking of the reference input (Blue Line in Fig. 8) of the closed-loop MFC control system-- that is, the TOC change in PSP. Besides intuitive observation (Fig.8), this error can be measured. The main evaluation indicators include: delay time, rise time, peak time, overshoot, settling time and steady-state error for tracking TOC changes in PSP: We have added the values of these parameters in the revision. It can be seen that the closed-loop MFC control system has high TOC tracking prediction accuracy.

4.More recent research should be cited ( the latest was from 2018) to show the related trends and novelty of this research.

This suggestion is quite reasonable. We have added some into the revised paper.

References:

Wang, Y., Xu, H., Wang, X., Gao, Y., Su, X., Qin, Y., Xing, L., Multi-Sub-Inlets at Cathode Flow-Field Plate for Current Density Homogenization and Enhancement of PEM Fuel Cells in Low Relative Humidity. Energy Conversion and Management, 2022, 252: 115069

Zhao T , Xie B , Yi Y , et al. Sequential flowing membrane-less microbial fuel cell using bioanode and biocathode as sensing elements for toxicity monitoring[J]. Bioresource Technology, 2019, 276:276-280.

Wang, Y., Wang, X.D., Qi, Y.Z., Zhang, L., Wang, Y.L. Three-dimensional numerical study of a cathode gas diffusion layer with a through/in plane synergetic gradient porosity distribution for PEM fuel cells. International Journal of Heat and Mass Transfer, 2022: 188, 122661.

Zang Y , Zhao T , Xie B , et al. A bio-electrochemical sensor based on suspended Shewanella oneidensis MR-1 for the sensitive assessment of water biotoxicity[J]. Sensors and Actuators B: Chemical, 2021, 341:130004.

Zang Y , Zhao H , Cao B , et al. Enhancing the sensitivity of water toxicity detection based on suspended Shewanella oneidensis MR-1 by reversing extracellular electron transfer direction[J]. Analytical and Bioanalytical Chemistry, 2022, 414(9):3057-3066.

Wang, Y., Xu, H., Zhang, Z., Li, H., Wang X. Lattice Boltzmann simulation of a gas diffusion layer with a gradient polytetrafluoroethylene distribution for a proton exchange membrane fuel cell. Applied Energy, 2022, 320:119248.

5.More information related to Fig. 5 should be provided. For example, how was ‘copy of MFC-biosensor’ made? What was ‘Mux’?

Thank you very much for your reminding.We have added relevant content in the revised manuscript, and introduced the modules in the simulation model in detail.

Reviewer 3 Report

This manuscript insisted that the microbial fuel cell sensor for the in-situ TOC monitoring in the public swimming pools. The reviewer thinks this manuscript is well-addressed, but the author did not consider the MFC condition. To the reviewer's knowledge, it is difficult to make the same condition of the biofilm on the electrode surface every single reactor operation. Therefore, if the biofilm condition may be changed, the equation is also much different. 

And the author knows that the microbial is not in stable condition and is inaccurate in practical application, but the author matched a single campaign data with a simulated result. in this case, the reviewer has doubts about the result's reliability. Thus, the reviewer suggests updating the result as duplicating or triplicating actual MFC operation data.  

And the other concern is below.

The author did not update the MFC condition clearly, such as inoculum, artificial substracts and pre-culture period for making the biofilm, etc. It is a critical factor in MFC cultivation. So, the author needs to update the material method and biofilm enrichment data in the result section.

Also, the author insists the microbial fuel cell can monitor other substances in water, such as heavy metal ions, chemical toxicants, and so forth, in the abstract. Accordingly, to the literature review, toxicity chemical detection is possible on the MFC, but it is difficult to recover the MFC after detection due to microbial damage. So, simulated data may not accurate in real applications. What does the author think? 

Author Response

Response to reviewer 3

We would like to thank the reviewers for the valuable comments and constructive suggestions that have improved the quality and clarity of the manuscript. Revised parts are marked in the manuscript.

This manuscript insisted that the microbial fuel cell sensor for the in-situ TOC monitoring in the public swimming pools. The reviewer thinks this manuscript is well-addressed, but the author did not consider the MFC condition. To the reviewer's knowledge, it is difficult to make the same condition of the biofilm on the electrode surface every single reactor operation. Therefore, if the biofilm condition may be changed, the equation is also much different.

Thank you for this good advice. As you pointed out, the environment of different biofilms varies. The kinetic model we identified describes the dynamic characteristics between TOC and electricity production, not for individual reactors, but for the whole reactor. The variation of biofilm conditions in a single reactor does not have much effect on the overall electrokinetic characteristics.

It is important to point out that the innovation in this paper is that we design the MFC as a closed-loop control system. According to the principle of cybernetics, the closed-loop control is highly resistant to disturbance. Under a certain limit of internal changes and external disturbances, as well as the inaccurate model, the feedback controller can also have a good compensation effect, offset their adverse effects, so that MFC has good dynamic prediction performance.

And the author knows that the microbial is not in stable condition and is inaccurate in practical application, but the author matched a single campaign data with a simulated result. in this case, the reviewer has doubts about the result's reliability. Thus, the reviewer suggests updating the result as duplicating or triplicating actual MFC operation data.

Thank you very much for your comments.

The MFC model we use for prediction is the kinetic equation -- transfer function, rather than the linear regression equation. As is known to all, the time domain function corresponding to the transfer function, namely, inverse Laplace transformation, is an ordinary differential equation model that can reflect the mechanism. Instead of black-box statistical regression equations, it reveals the kinetic mechanism of the effect of TOC on the power generation performance of MFC. Therefore, the transfer function tracks the process of change, not the final stable state. So, different from the regression equation, it does not need to be repeated many times. Instead, it makes a long time observation on the electricity generation of MFC to obtain the time series data about the input and output. To identify the dynamic model between the TOC variation of MFC and the electrical performance of MFC.

Unstable state, we said in the article is aimed at MFC internal flora in terms of internal and external environment condition, thanks to the comments of the reviewer, we established in revised manuscript adds MFC model premise, that is in the operation process of MFC, flora in stable state, even if the disturbance. And the community can return to its original stable state. In PSP, this condition is guaranteed.

Again, it is important to point out that the innovation in this paper is that we design the MFC as a closed-loop control system. According to the principle of cybernetics, the closed-loop control is highly resistant to disturbance. Under a certain limit of internal changes and external disturbances, the feedback controller can also well compensate the prediction ability of the MFC, offset their adverse effects, and make the MFC have good dynamic prediction performance.

And the other concern is below.

The author did not update the MFC condition clearly, such as inoculum, artificial substrates and pre-culture period for making the biofilm, etc. It is a critical factor in MFC cultivation. So, the author needs to update the material method and biofilm enrichment data in the result section.

This suggestion is quite reasonable. 

Three groups of dual-chamber MFC were constructed, in which the anode and cathode compartments were 25 mL and 42 mL, respectively. The two chambers were separated by a proton exchange membrane (Nafion 117, Hesen Electric Corporation, China). The anode was 3 cm×2.5 cm carbon cloth (HCP330, Hesen Electric Corporation, China), which was soaked in acetone overnight before use, then dried and subjected to high temperature ammonification. The cathode was 2.0 cm×2.0 cm platinum-loaded carbon paper with a platinum-loaded capacity of 0.5 mg/cm2. The external resistance of the MFC load is 330 Ω. The anode inoculation source was the anode effluent from MFC that had been running stably for more than half a year in the laboratory, and the electrochemically active microorganism of the mixed strains was mainly Geobacter. After inoculation, the MFC of each group was placed in a constant temperature incubator at 27.0±0.5 â„ƒ, and the anode chamber was self-circulated at 3 mL/min through a peristaltic pump to accelerate mass transfer and promote the formation of biofilms. Each MFC was connected to the data acquisition system to measure the output voltage of the MFC in real time. When the maximum voltage does not increase and three cycles of repeatable output voltage are detected, the MFC startup is considered to be complete.

Anolyte and catholyte were prepared prior to use. The fresh anolyte used in this study consisted of 5.85 g NaCl, 0.13 g KCl, 0.31 g NH4Cl, 6.08 g NaH2PO4•2H2O, 21.83 g Na2HPO4•12H2O, 12.50 mL trace minerals solution and 5.00 mL vitamin solution in 1.00 L deionized water. The catholyte consisted of 5.85 g NaCl, 6.08 g NaH2PO4•2H2O and 21.83 g Na2HPO4•12H2O in 1.00 L deionized water. During the start-up process, sodium acetate (NaAc) was added to fresh anolyte as carbon source to achieve a final concentration of 0.82 g/L.

References:

W.E. Balch, G.E. Fox, L.J. Magrum, C.R. Woese, R.S. Wolfe, Methanogens: reevaluation of a unique biological group, Microbiol. Rev. 43 (1979) 260–296.

  1. Jiang, P. Liang, P.P. Liu, Y.H. Bian, B. Miao, X.L. Sun, H.L. Zhang, X. Huang, Enhancing signal output and avoiding BOD/toxicity combined shock interference by operating a microbial fuel cell sensor with an optimized background concentration of organic matter, Int. J. Mol. Sci. 17 (2016) 1392–1399.

Yi Y , Xie B , Zhao T , et al. Effect of external resistance on the sensitivity of microbial fuel cell biosensor for detection of different types of pollutants[J]. Bioelectrochemistry, 2018, 125:71-78.

Also, the author insists the microbial fuel cell can monitor other substances in water, such as heavy metal ions, chemical toxicants, and so forth, in the abstract. Accordingly, to the literature review, toxicity chemical detection is possible on the MFC, but it is difficult to recover the MFC after detection due to microbial damage. So, simulated data may not accurate in real applications. What does the author think?

Thanks for your comments. In this study, the range of TOC concentration in PSP was measured on the premise that it would not cause significant damage to the microbial community. In the monitoring of heavy metal ions and chemical toxicants, if there is great damage to the microbial community structure, the microbial community will recover by itself to a certain extent, which can ensure the ideal accuracy within a certain measurement range. However, at this time, the electricity generation process of MFC will produce strong nonlinear dynamic characteristics. In this case, we can use nonlinear dynamic models with higher fault tolerance, such as fuzzy inference system (FIS) and artificial neural network (ANN) to describe it. The closed-loop control system can be designed by such as robust control, self-adaptive control, fuzzy logic control, artificial neural networks and other advanced cybernetics methods to complete dynamic forecasts from heavy metal ions, chemical toxicants. These contents have been described in the Discussion section.

Round 2

Reviewer 2 Report

The revised manuscript is the same as the previously submitted manuscript.

Author Response

Response to reviewer 2

The revised manuscript is the same as the previously submitted manuscript.

Thanks for your comment. I think it is a misunderstanding for the revised manuscript (Round 1), you may not find the revised version in the review system. To prevent such things happening, the response to reviewer (Round 1) and the revised manuscript was attached to the end.

And you can also find the new version in the review system.

Reviewer 3 Report

This manuscript is well addressed in the revised version.

And, the Author should double-check the error word or style before publication.

Author Response

Response to reviewer 3

We would like to thank the reviewers for the valuable comments and constructive suggestions that have improved the quality and clarity of the manuscript. Revised parts are marked in the manuscript.

This manuscript is well addressed in the revised version.

And, the Author should double-check the error word or style before publication.

Thank you very much for your positive and valuable comments. We tried our best to avoid error words and check the style in the revised manuscript.

Thanks again!
